# THE INTRIGUING ROLE OF MODULE CRITICALITY IN THE GENERALIZATION OF DEEP NETWORKS

**Niladri S. Chatterji**
University of California, Berkeley
chatterji@berkeley.edu

**Behnam Neyshabur**
Google
neyshabur@google.com

**Hanie Sedghi**
Google
hsedghi@google.com

## ABSTRACT

We study the phenomenon that some modules of deep neural networks (DNNs) are more *critical* than others. Meaning that rewinding their parameter values back to initialization, while keeping other modules fixed at the trained parameters, results in a large drop in the network's performance. Our analysis reveals interesting properties of the loss landscape which leads us to propose a complexity measure, called *module criticality*, based on the shape of the valleys that connect the initial and final values of the module parameters. We formulate how generalization relates to the module criticality, and show that this measure is able to explain the superior generalization performance of some architectures over others, whereas, earlier measures fail to do so.

## 1 INTRODUCTION

Neural networks have had tremendous practical impact in various domains such as revolutionizing many tasks in computer vision, speech and natural language processing. However, many aspects of their design and analysis have remained mysterious to this date. One of the most important questions is "what makes an architecture work better than others given a specific task?" Extensive research in this area has led to many potential explanations on why some types of architectures have better performance; however, we lack a unified view that provides a complete and satisfactory answer. In order to attain a unified view on superiority of one architecture over another in terms of generalization performance, we need to come up with a measure that effectively captures this.

Analyzing the generalization behavior of neural networks has been an active area of research since Baum & Haussler (1989). Many generalization bounds and complexity measures have been proposed so far. Bartlett (1998) emphasized the importance of the norm of the weights in predicting the generalization error. Since then various analysis have been proposed. These results are either based on covering number and Rademacher complexity (Neyshabur et al., 2015; Bartlett et al., 2017; Neyshabur et al., 2019; Long & Sedghi, 2019; Wei & Ma, 2019), or they use approaches similar to PAC-Bayes (McAllester, 1999; Dziugaite & Roy, 2017; Neyshabur et al., 2017; 2018; Arora et al., 2018; Nagarajan & Kolter, 2019a; Zhou et al., 2019). Recently authors have emphasized on the role of distance to initialization rather than norm of the weights in generalization (Dziugaite & Roy, 2017; Nagarajan & Kolter, 2019b; Neyshabur et al., 2019; Long & Sedghi, 2019). Earlier results have an exponential dependency on the depth and focus on fully connected networks. More recently, Long & Sedghi (2019) provided generalization bounds for convolutional neural networks (CNNs) and fully connected networks used in practice and their bounds have linear dependency on the depth.

Despite the success of earlier works in capturing the dependency of generalization performance of a model on different parameters, they fail at the following task: Rank the generalization performance of candidate architectures for a specific task such that the ranking aligns well with the ground truth. Moreover, majority of these bounds are proposed for fully connected modules and it is not straightforward to evaluate them for different architectures such as ResNets.

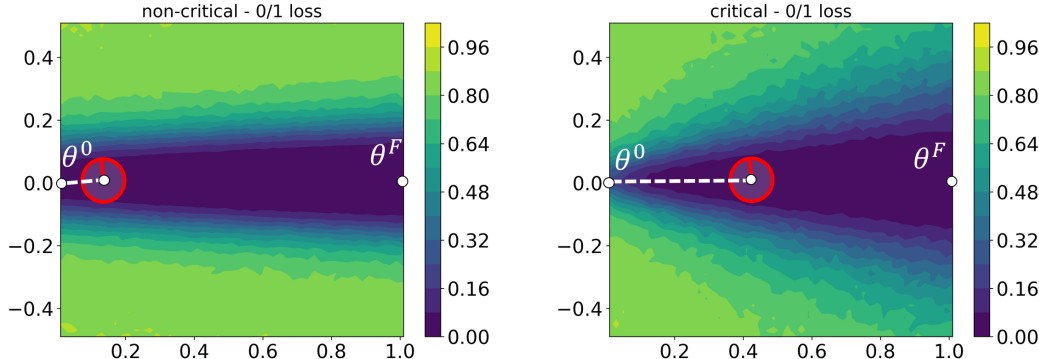

Figure 1: **Module Criticality**: Loss values in the valleys that connect the initial weights $\theta^0$ to the final weights $\theta^F$ of a non-critical (left) and a critical (right) module in the ResNet18 architecture. Given a ball with radius $r$ (length of the red line), module criticality can be defined as how far one can push the ball in the valley towards initialization (length of the white dashed line) divided by the radius $r$. Hence, non-critical modules are the ones with a wide valley connecting the initial weight vector to the final one whereas in critical modules, the valley either becomes too sharp or the loss values start to increase when the ball comes too close to the initial weight. The $x$ axis is simply chosen to be parallel to $\theta^F - \theta^0$ and the $y$ axis is a compact representation of all other dimensions generated by adding Gaussian noise to the points on the convex combination of $\theta^0$ and $\theta^F$ and evaluating the loss. The sign on the $y$ axis is decided based on the sign of the inner product of the noise to $\theta^0$.

Every DNN architecture is a computation graph where each node is a module[1]. We are interested in understanding how different modules in the network interact with each other and influence generalization performance as a whole. To do so, we delve deeper into the phenomenon of "module criticality" which was reported by Zhang et al. (2019a). They observed that modules of the network present different robustness characteristics to parameter perturbation. Specifically, they look into the following perturbation: Rewind one module back to its initialization value while keeping all other modules fixed (at the final trained value). They note that the impact of this perturbation on network performance varies between modules and depends on which module was rewound. Some modules are "critical" meaning that rewinding their value to the initialization harms the network performance, while for others the impact of this perturbation on performance is negligible. They show that various conventional DNN architectures exhibit this phenomenon.

Let us now informally define what we mean by the measure "module criticality" (see Figure 1). For each module, we move on a line from its final trained value to its initialization value (convex combination[2] path) while keeping all other modules fixed at their trained value. Then we measure the performance drop. Let $\theta_i^\alpha = (1-\alpha)\theta_i^0 + \alpha\theta_i^F, \alpha \in [0,1]$ be the convex combination between initial weights $\theta_i^0$ and the final weights $\theta_i^F$ at module $i$, where $\alpha_i$ is the minimum value between 0 and 1 when performance (train error) of the network drops by at most a threshold value $\epsilon$. If $\alpha_i$ is small we can move a long way back to initialization without hurting performance and the "module criticality" of this module would be low. Further, we also wish to incorporate the robustness to noise (that is, the valley width) for the module along this path. If the module is robust to noise along this path (that is, the valley is wide) then the module criticality would again be low (see Definition 3.1 for a formal definition).

In this paper, we seek to study this phenomenon in depth and shed some light on it by showing that conventional complexity measures cannot capture criticality (see Section 2). Next, we theoretically formulate this phenomenon and analyze its role in generalization. Through this analysis, we provide

---

[1]A module is a node in the computation graph that has incoming edges from other modules and outgoing edges to other nodes and performs a linear transformation on its inputs. For layered model such as VGG, module definition is equivalent to definition of a layer.

[2]A convex combination of two points is a linear combination of them where the coefficients are non-negative and sum to 1. Every convex combination of two points lies on the line segment between the two.

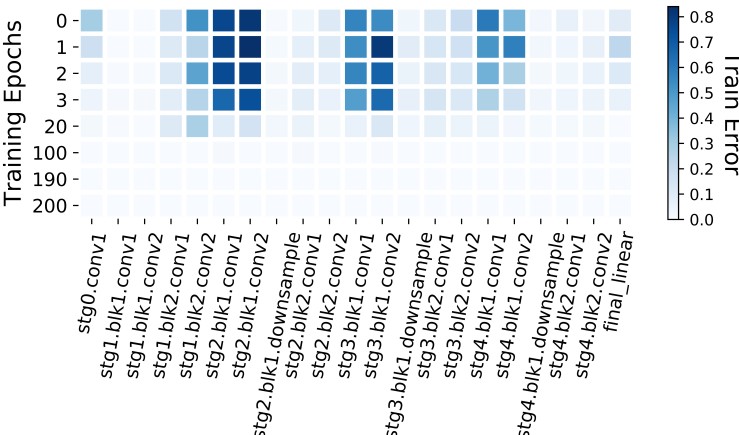

Figure 2: Analysis of rewinding modules to initialization for the ResNet-18 architecture. Each row represents a module in ResNet18-v1 and each column represents a particular training epoch to which this module is rewound to. The difference from analysis of Zhang et al. (2019a) is that we rewind each module, whereas Zhang et al. (2019a) rewind the entire ResNet blocks.

a new generalization measure that captures the dissimilarity of different modules and depicts how it influences the generalization of the corresponding DNN. Intuitively, the closer we can get to initialization for each module, the better the generalization.

We analyze the relation between generalization and module criticality through a PAC-Bayesian analysis. In Section 3 we show that it is the overall network criticality measure and not the number of critical modules that controls generalization. If the network criticality measure is smaller for an architecture, it has better generalization performance. In Section 4, we demonstrate through various experiments that our proposed measure is able to distinguish between different network architectures in terms of their generalization performance. Moreover, the network criticality measure is able to correctly rank the generalization performance of different architectures better than the measures proposed earlier.

**Notation:** We use upper case letters for matrices. The operator norm and Frobenius norm of $M$ are denoted by $\|M\|_2$, $\|M\|_{\mathrm{Fr}}$ respectively. For $n \in \mathbb{N}$, we use $[n]$ to denote the set $\{1, \ldots, n\}$. Let $\mathcal{L}_S(f)$ be the loss of function $f$ on the training set $S$ with $m$ samples. We are mainly interested in the classification task where $\mathcal{L}_S(f) = \frac{1}{m} \sum_{(x,y) \in S} \mathbf{1}[f(x)[y] \leq \max_{j \neq y} f(x)[j]]$. For any $\gamma > 0$, we also define margin loss $\mathcal{L}_{S,\gamma}(f) = \frac{1}{m} \sum_{(x,y) \in S} \mathbf{1}[f(x)[y] \leq \gamma + \max_{j \neq y} f(x)[j]]$. Let $\mathcal{L}_D(f)$ be the loss of function $f$ on population data distribution $D$ defined similar to $\mathcal{L}_S(f)$. We will denote the function parameterized by $\Theta$ by $f_\Theta$.

## 2 TOWARDS UNDERSTANDING MODULE CRITICALITY

### 2.1 SETTING

A DNN architecture is a directed acyclic computation graph which may or may not be layered. In order to have a unifying definition between different architectures, we use the notion of a "*module*". A module is a node in the computation graph that has incoming edges from other modules and outgoing edges to other nodes, and performs a linear transformation on its inputs. For a layered model such as a VGG, a module is equivalent to a layer. On the other hand, in a ResNet some modules are parallel to each other. For example, a downsample module and the concatenation of two convolutional modules in a ResNet18-v1 architecture. Note that, similar to conventional definitions the non-linearity (such as a ReLU) is not part of the module.

Let $\Theta = (\theta_1, \ldots, \theta_d)$ correspond to all parameters of a DNN with $d$ modules, where $\theta_i$ refers to the weight matrix (or operator matrix in case of convolution) at module $i$ and $\theta_i^0, \theta_i^F$ refer to the value

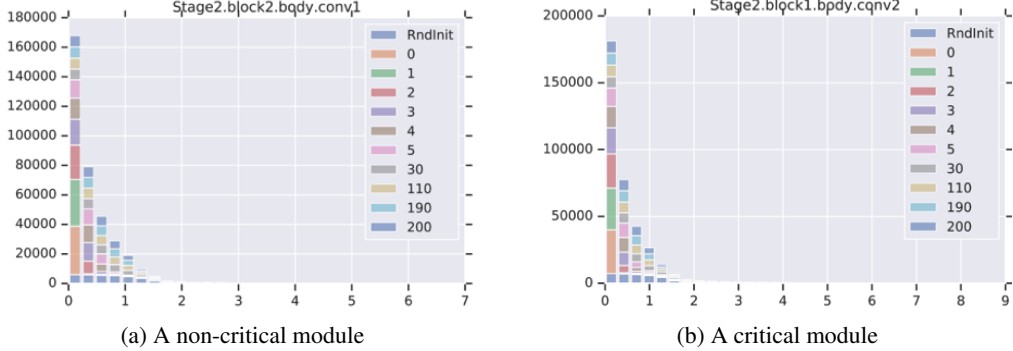

(a) A non-critical module            (b) A critical module

Figure 3: Spectrum of a non-critical and a critical module during different epochs of training.

of weight matrix at initialization and the end of training, respectively. For sequential architectures, $d$ is equal to the depth of the network but that is not necessarily true for a general architectures such as a ResNet.

## 2.2 ROBUSTNESS TO REWINDING

Consider the following perturbation to a trained network at some training epoch as considered by Zhang et al. (2019a). For each module in the network, rewind its value back to its value at this training epoch while keeping the values of all other modules fixed (at their final trained value). Next, measure the change in performance of the model before and after this manipulation. We repeat a similar analysis that differs from that of Zhang et al. (2019a) in one detail. Zhang et al. (2019a) rewind the whole ResNet block at once, whereas, we rewind each module (each convolutional module) separately. This rewind analysis is shown in Figure 2 for ResNet18-v1. Each column represents a module in ResNet18-v1 and each row represents a particular training epoch to which this module is rewound to. Similar to earlier analysis, we observe that for many modules of the network, this manipulation does not influence the network performance drastically, while, for some others the impact is more pronounced. For example, in Figure 2 we look at the effect of rewinding on train error. The "Stage2.block1.conv2" module is critical, whereas, most other modules, once rewound, do not affect the performance. In Figure 6 in Appendix E we plot the effect of rewinding on different performance criteria (train loss, train error and test error) and observe that they exhibit a similar trend.

**A stable phenomena:** The plots in Figure 2 capture a network trained with SGD with weights initialized using the standard Kaiming initialization (He et al., 2015). To ensure that the observed phenomenon is not an artifact of the training method and the initialization scheme, we repeated the experiments with different initialization and optimization methods. We saw a similar pattern. For example, Figures 7a, 7b in the appendix illustrate the pattern when we changed the initialization to Fixup (Zhang et al., 2019b), and when we replace SGD with Adam (Kingma & Ba, 2014) respectively.

## 2.3 WHAT MEASURES FAIL TO DISTINGUISH CRITICAL LAYERS

**Spectrum of weight matrices:** We explore the change in the spectrum of different weight matrices on rewinding and note that the spectrum for a critical and non-critical module look similar. This is shown in Figure 3. We calculate the spectrum of convolutional layers using the algorithm by Sedghi et al. (2019).

**Distance to initialization:** Next, we analyze the operator norm of difference from initialization for each module. Figure 8 in the appendix depicts this and reveals no difference between critical and non-critical modules. A similar plot was explored by Zhang et al. (2019a), where they find that the Frobenius norm and the infinity norm also fail to capture criticality.

**Change in the activation patterns:** We investigated the change in the activation patterns of a network when we rewind a module. To do this, we study the similarity between two networks: 1. The original trained network and 2. The network with a rewound module. We use CKA (Kornblith et al., 2019) as the measure of similarity. For a non-critical module, the original and rewound networks are similar and in case of a critical module, the similarity between the activation patterns between the two networks degrades gradually rather than abruptly. See Figure 9 in Appendix E.

## 3 GENERALIZATION BOUNDS BASED ON MODULE CRITICALITY

Our goal is to understand criticality and how it affects the generalization performance of a DNN. Inspired by the rewind to initialization experiments of Zhang et al. (2019a), we take one step further and consider changing the value of each module, to the convex combination of its initial and final value. That is, for each module $i$, we replace $\theta_i$ with $\theta_i^\alpha = (1-\alpha)\theta_i^0 + \alpha\theta_i^F, \alpha \in [0, 1]$, and keep all other layers fixed. Then we look at the effect of this perturbation on the performance of the network.

Figure 4 depicts how the train error, test error and train loss change as we decrease the value of $\alpha$ in $\theta_i^\alpha$ when $i$ refers to a critical module (yellow dashdot curve), a non-critical module (red dashed curve) and all modules (blue solid curve). We find that along this convex combination path all these performance measures degrade monotonically (increase in error and loss), as we move from the final weights to the initial weights.

The above experiment shows the effect of moving along a convex combination between module's initial and trained value. To capture the relation between criticality and generalization, we are interested in also accounting for the width of the valley as we move from the final value to the initial value. In particular, we are interested in analyzing what happens if we are moving inside a ball of some radius $\sigma_i$ around each point in this path. PAC-Bayesian analysis, looks for a ball around final value of parameters such that the loss does not change if we move in this ball. Bringing this idea together with the one mentioned above, we are interested in moving from the final value to the initial value in a valley of some radius, and want to find out how far we can move on this path. Intuitively, being able to move closer to initialization values indicate that the effective function class is smaller and hence the network should generalize better. For example, in the extreme case where none of the weights change from their initialization value the function class would be a single function (the initial function) and the generalization error would be very low ($\sim$0%) as both the train and test error would be very high (but equal). In this paper, we consider the case of trained models where train error is very low and hence low generalization error means good performance on test data.

**Definition 3.1** (Module and Network Criticality). *Given an $\epsilon > 0$ and network $f_\Theta$, we define the module criticality for module $i$ as follows:*

$$\mu_{i,\epsilon}(f_\Theta) = \min_{0 \le \alpha_i, \sigma_i \le 1} \left\{ \frac{\alpha_i^2 \left\| \theta_i^F - \theta_i^0 \right\|_{\text{Fr}}^2}{\sigma_i^2} : \mathbb{E}_{u \sim \mathcal{N}(0, \sigma_i^2)}[\mathcal{L}_S(f_{\theta_i^\alpha + u, \Theta_{-i}^F})] \le \epsilon \right\}, \qquad (1)$$

*We also define the network criticality as the sum of the module criticality over modules of the network:*

$$\mu_\epsilon(f_\Theta) = \sum_{i=1}^{d} \mu_{i,\epsilon}(f_\Theta). \qquad (2)$$

Here, $\mathcal{L}_S$ denotes the empirical zero-one loss over the training set, $f_{\theta_i^\alpha, \Theta_{-i}^F}$ is the DNN's function value where weight matrix corresponding to $i^{th}$ module is replaced by $\theta_i^\alpha$ and all other modules are fixed at their values in the end of training, $\Theta_{-i}^F$. $\theta_i^\alpha = (1-\alpha)\theta_i^0 + \alpha\theta_i^F$, where $\theta_i^0$ is the value of the weight matrix at initialization and $\theta_i^F$ is the trained value.

Intuitively, network criticality measure is sum of module criticalities. This is also theoretically derived using the analysis below.

### 3.1 A PAC-BAYESIAN GENERALIZATION BOUND

We attempt to understand the relationship between module criticality, and generalization by deriving a generalization bound using the PAC-Bayesian framework (McAllester, 1999). Given a prior

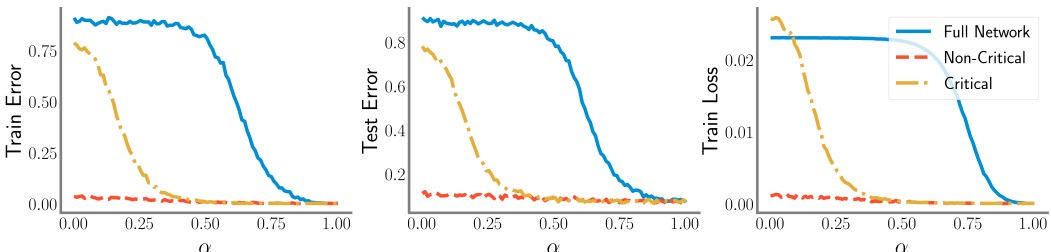

Figure 4: Performance degradation as we move on convex combination path from final to initial value of modules. We find that along this path the training error (as well as test error and train loss) increases monotonically from the final weights to initial weights. The blue (solid) curve is when we replace all the parameters in the network by the convex combination between their initial and final value simultaneously, the red (dashed) curve corresponds to moving on the convex path for a single (non-critical) layer and the yellow (dashdot) curve corresponds to moving on the convex path for a critical layer in ResNet-18 architecture.

distribution over the parameters that is picked in advance before observing a training set, a posterior distribution over the parameters that could depend on the training set and a learning algorithm, the PAC-Bayesian framework bounds the generalization error in terms of the Kullback-Leibler (KL) divergence (Kullback & Leibler, 1951) between the posterior and the prior distribution. We use PAC-Bayesian bounds as they hold for any architecture.

The intuition from Figure 4 suggests moving the parameters of each module as close as possible to the initialization value before harming the performance. For such $\alpha_i$, we can then define the posterior $Q_i$ for module $i$ to be a Gaussian distribution centered at $\theta_i^\alpha$ with covariance matrix $\sigma_i^2 I$, that is, as if we have additive noise $u_i \sim \mathcal{N}(0, \sigma_i I)$. We use $\Theta^\alpha$ to refer to the case where all the parameters $\theta_i$ are replaced with $\theta_i^{\alpha_i}$ and matrix $U$ includes all the noise $u_i$. Then the following theorem holds.

**Theorem 3.2.** *For any data distribution $D$, number of samples $m \in \mathbb{N}$, for any $0 < \delta < 1$, for any $0 < \sigma_i \leq 1$ and any $0 \leq \alpha_i \leq 1$, with probability $1 - \delta$ over the choice of the training set $S_m \sim D$ the following generalization bound holds:*

$$\mathbb{E}_U[\mathcal{L}_D(f_{\Theta^\alpha + U})] \leq \mathbb{E}_U[\mathcal{L}_S(f_{\Theta^\alpha + U})] + \sqrt{\frac{\frac{1}{4}\sum_{i=1}^d k_i \log\left(1 + \frac{\alpha_i^2 \|\theta_i^F - \theta_i^0\|_{\text{Fr}}^2}{k_i \sigma_i^2}\right) + \log\left(\frac{m}{\delta}\right) + \widetilde{\mathcal{O}}(1)}{m - 1}},$$

*where $k_i$ is the number of parameters in module $i$. For example, for a convolution module with kernel size $q_i \times q_i$ and number of output channels $c_i$, $k_i = q_i^2 c_{i-1} c_i$.*

The exact bound including the constants and the proof of the theorem above is given in Appendix A. Theorem 3.2 already gives us some insight into generalization of the original network. However, it is not exactly a generalization bound on the original network but rather on a perturbed network. We conjecture that for almost any realistic distribution $D$, any random $\Theta^0$, any $\Theta^F$ achieved by known gradient based optimization algorithms, any $0 \leq \alpha \leq 1$ and any $\sigma \geq 0$, the test error does not improve by taking a convex combination of parameters and their initial values followed by Gaussian perturbation. Therefore, we have that $\mathcal{L}_D(f_{\Theta^F}) \leq \mathbb{E}_U[L_D(f_{\Theta^\alpha + U})]$. The following corollary restates Theorem 3.2 by using this assumption and optimizing over $\alpha$ and $\sigma$ in the bound.

**Corollary 3.3.** *For any data distribution $D$, number of samples $m \in \mathbb{N}$. For any $\epsilon > 0$, for any $0 < \delta$, if $\mathcal{L}_D(f_{\Theta^F}) \leq \mathbb{E}_U[\mathcal{L}_D(f_{\Theta^\alpha + U})]$ where $u_i \sim \mathcal{N}(0, \sigma_i I)$, then with probability $1 - \delta$ over the choice of the training set $S_m \sim D$, the following generalization bound holds*

$$\mathcal{L}_D(f_\Theta) \leq \epsilon + \sqrt{\frac{\frac{1}{4}\mu'_\epsilon(f_\Theta) + \log\left(\frac{m}{\delta}\right) + \widetilde{\mathcal{O}}(1)}{m - 1}},$$

*where $\mu'_\epsilon(f_\Theta)$ is calculated as follows:*

$$\mu'_\epsilon(f_\Theta) = \min_{0 \leq \alpha, \sigma \leq 1}\left\{\sum_i \frac{\alpha_i^2 \|\theta_i^F - \theta_i^0\|_{\text{Fr}}^2}{\sigma_i^2} : \mathbb{E}_U[\mathcal{L}_S(f_{\Theta^\alpha + U})] \leq \epsilon\right\}.$$

Note that the above bound uses a slightly different notion of network criticality compared to Definition 3.1, as the bound requires finding $\alpha$ and $\sigma$ values simultaneously for all modules, whereas, Definition 3.1 allows us to decouple the search over $\alpha$ and $\sigma$.

**Deterministic generalization bound for convolutional networks**  Although PAC-Bayesian bounds are data-dependent and hence numerically superior, they provide less insight about the underlying reason that results in generalization. For example, the flatness of the solution after adding Gaussian perturbation can be computed numerically. But computing this value does not reveal what properties of the network enforce the loss surface around a point to be flat. On the other hand, deterministic norm-based generalization bounds are numerically much looser yet they provide better insights into the dependence of generalization on different network parameters. In Appendix B, we build on the results of Theorem 3.2 to present a norm-based deterministic bound using module criticality.

In this section, we intuitively justified network criticality measure and related the generalization of a DNN to the network criticality measure in Corollary 3.3. In the next section, we empirically show that the network criticality measure is able to correctly rank the generalization performance of different architectures better than measures proposed earlier.

## 4    EXPERIMENTS

We perform several experiments to compare our network criticality measure to earlier complexity measures in the literature. Our experiments are performed on the CIFAR10 and CIFAR100 datasets. For all experiments, implementation and architecture details are presented in Appendix C.

Table 1 summaries the quantities that are calculated in this section. The quantity SoSP was proposed by Long & Sedghi (2019). For the last two measures, we calculate $\sigma_i$ and $\alpha_i$ as per Definition 3.1. In our experiments, each module is a single convolutional or linear layer. This represents a natural choice where each module is a linear transformation (with respect to the parameters). This choice also leads to the lowest number of modules such that each module is a linear transformation.

Table 1: Quantities of Interest

| | |
|---|---|
| Generalization Error (GE) | $\mathcal{L}_D(f_\Theta) - \mathcal{L}_S(f_\Theta)$ |
| Product of Frobenius Norms (PFN) | $\Pi_i \|\theta_i^F\|_{\text{Fr}}$ |
| Product of Spectral Norms (PSN) | $\Pi_i \|\theta_i^F\|_2$ |
| Distance to Initialization (DtI) | $\sum_i \|\theta_i^0 - \theta_i^F\|_{\text{Fr}}^2$ |
| Number of Parameters (NoP) | Total number of parameters in the network |
| Sum of Spectral Norms (SoSP) | Total number of parameters $\times(\sum_i \|\theta_i^0 - \theta_i^F\|_2)$ |
| PAC Bayes (at error threshold 0.1) | $\sum_i \|\theta_i^0 - \theta_i^F\|_{\text{Fr}}^2/\sigma_i^2$ |
| Network Criticality Measure (at error threshold 0.1) | $\sum_i \alpha_i^2 \|\theta_i^0 - \theta_i^F\|_{\text{Fr}}^2/\sigma_i^2$ |

First, as a sanity check we use our complexity measure (lower is better) to compare between a ResNet18 trained on true labels and a ResNet18 trained on data where $20\%$ of the labels are randomly corrupted. As seen in Figure 5a, our measure is able to correctly capture that the network trained with true label generalizes better than the one trained on corrupted labels (4.62% error vs. 35% error).

Next, in Table 2 we compare the generalization performance of several conventional DNN architectures trained on the CIFAR10 dataset. There is a particular ranking of the networks based on their generalization error and it is desirable for a complexity measure to capture this ranking. Therefore, we compare the rankings proposed by network criticality measure and complexity measures from the literature with the empirical rankings obtained in the experiment. To do this, we calculate the *Kendall's $\tau$ correlation coefficient* (Kendall, 1938) which is defined as follows:

$$\text{Kendall's } \tau = \frac{\text{\# of pairs where the rankings agree} - \text{\# of pairs where the rankings disagree}}{\text{\# pairs}}.$$

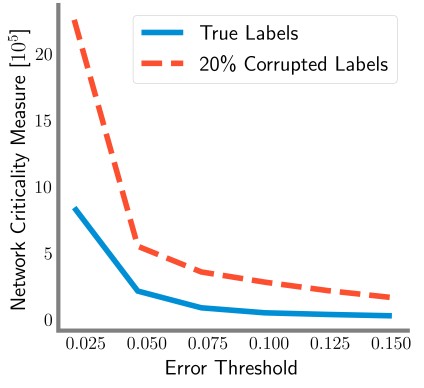

(a) Comparing ResNet18 on trained true labels vs. corrupted labels

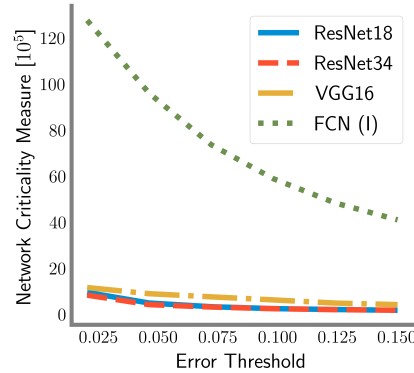

(b) Comparing ResNet18, ResNet34, VGG16 and FCN (I).

Figure 5: Network criticality as a function of error threshold for networks trained on CIFAR10.

This coefficient lies between $-1$ and 1, where 1 denotes a high correlation between the two set of rankings. Table 2 shows that the Kendall's $\tau$ coefficient between our network criticality measure and the generalization error is higher than all other complexity measures that we compared to.

We find that our measure correctly ranks the generalization performance of the networks – ResNet18, ResNet101, VGG16 and FCN (I). It fails to correctly identify the correct rank of ResNet34, ResNet50, DenseNet121, VGG11 and FCN (II).

We also repeat the above experiment for the same networks trained on the CIFAR100 dataset (see Table 3). We note that network criticality measure correctly predicts the ranking of generalization performance for ResNet101, ResNet34, ResNet18, ResNet50, VGG16 and the 3-layer fully connected networks (FCN). We find that the generalization error of ResNet101 is the lowest which is correctly captured only by our complexity measure and not by any other measure. Further, the Kendall's $\tau$ correlation coefficient between the ranking based of the generalization error and our network criticality measure is 0.55 which is again higher than this coefficient for any other complexity measure. Our measure only fails to capture the ranking of VGG11 and DenseNet121 relative to the other DNN architectures and the relative ranking between FCN (I) and FCN (II).

Table 2: Measuring complexity of different architectures trained on CIFAR10.

| Network | GE | PFN | PSN | DtI | NoP | SoSP | PAC Bayes | Net. Criticality |
|---|---|---|---|---|---|---|---|---|
| ResNet18 | 4.61% | 1e22 | 4e14 | 3430 | 1.1e7 | 1.3e9 | 6.9e5 | 2.2e5 |
| ResNet34 | 6.3% | 2e37 | 3e24 | 4768 | 2.1e7 | 3.7e9 | 9.1e5 | 1.7e5 |
| ResNet50 | 6.6% | 4e56 | 4e20 | 10018 | 2.3e7 | 3.3e9 | 1.6e6 | 1.8e5 |
| ResNet101 | 6.4% | 8e110 | 3e32 | 18730 | 4.2e7 | 9.8e9 | 2.8e6 | 6.3e5 |
| DenseNet121 | 7.8% | 2e129 | 7e42 | 21359 | 6.8e6 | 2.0e9 | 1.2e6 | 4.1e5 |
| VGG11 | 8.51% | 1e11 | 1e6 | 2106 | 2.8e7 | 1.3e9 | 1.0e6 | 2.8e5 |
| VGG16 | 7.47% | 5e15 | 2e8 | 2341 | 3.4e7 | 2.1e9 | 1.2e6 | 2.70e5 |
| FCN (I) | 29.83% | 3e20 | 2e7 | 75221 | 2.0e7 | 4.6e8 | 9.0e6 | 5.7e6 |
| FCN (II) | 26.45% | 3e21 | 1e7 | 81258 | 5.0e7 | 2.0e9 | 9.5e6 | 6.2e6 |
| Kendall's $\tau$ | - | -0.22 | -0.33 | 0.38 | 0.16 | -0.53 | 0.42 | 0.55 |

We also perform an additional set of experiments in Appendix D where we calculate these complexity measures on four ResNet18 networks with changing channel widths.

## 5 CONCLUSION

In this paper, we studied the module criticality phenomenon and proposed a complexity measure based on module criticality that is able to correctly predict the superior performance of some DNN

Table 3: Measuring complexity of different architectures trained on CIFAR100.

| Network | GE | PFN | PSN | DtI | NoP | SoSP | PAC Bayes | Net. Criticality |
|---|---|---|---|---|---|---|---|---|
| ResNet18 | 30.6% | 2e22 | 9e14 | 4855 | 1.1e7 | 1.4e9 | 3.4e6 | 1.6e6 |
| ResNet34 | 29.3% | 1e37 | 1e22 | 6017 | 2.1e7 | 3.6e9 | 6.7e6 | 1.4e6 |
| ResNet50 | 31.1% | 7e57 | 9e23 | 12715 | 2.3e7 | 4.1e9 | 5.8e6 | 1.9e6 |
| ResNet101 | 25.5% | 2e112 | 5e36 | 21233 | 4.2e7 | 1.1e10 | 6.4e6 | 1.3e6 |
| DenseNet121 | 35.3% | 1e131 | 1e49 | 23702 | 6.9e6 | 2.4e9 | 4.5e6 | 2.6e6 |
| VGG11 | 43.5% | 6e13 | 1e8 | 5059 | 2.8e7 | 1.7e9 | 3.6e6 | 2.4e5 |
| VGG16 | 32.69% | 8e19 | 4e12 | 7010 | 3.4e7 | 3.6e9 | 8.1e6 | 4.6e6 |
| FCN (I) | 57.59% | 8e27 | 1e10 | 246636 | 2.0e7 | 8.3e8 | 2.4e7 | 1.2e7 |
| FCN (II) | 53.17% | 1e29 | 1e10 | 329296 | 5.0e7 | 3.7e9 | 3.3e7 | 2.3e7 |
| Kendall's $\tau$ | - | -0.27 | -0.47 | 0.33 | 0 | -0.42 | 0.22 | 0.55 |

architectures over others, for a specific task. We believe module criticality can be used as a road-map for designing new task-specific architectures. Proposing new regularizers that improve generalization performance by bounding criticality or spreading it among various modules of the network is an exciting direction for future work. Our measure could also be potentially used in architecture search where we could calculate this score over the training set to select architectures that generalize well on the unseen test set.

ACKNOWLEDGEMENTS

We would like to thank Samy Bengio and Chiyuan Zhang for valuable conversations, and Yann Dauphin for his help with the implementation of Fixup initialization. We would also like to thank Chiyuan Zhang for sharing the code for the paper "Are all layers created equal?" Part of this work was performed while the author NC was an intern at Google AI, Brain team.

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

# A  Proof of Thereorm 3.2

We start by stating the PAC-Bayes theorem which bounds the generalization error of any posterior distribution $Q$ on parameters $\Theta$ that can be reached using the training set given a prior distribution $P$ on parameters that should be chosen in advance and before observing the training set. Throughout this section given two scalar $p, q \in [0, 1]$ let $KL(p||q)$ denote the KL divergence between two Bernoulli distributions with success probabilities $p$ and $q$ respectively.

**Theorem A.1** (McAllester (1999)). *For any data distribution $D$, number of samples $m \in \mathbb{N}$, training set $S_m \sim D$, and prior distribution $P$ on parameters $\Theta$, posterior distribution $Q$, for any $0 < \delta$, with probability $1 - \delta$ over the draw of training data we have that*

$$KL\left(\mathbb{E}_{\Theta \sim Q}[\mathcal{L}_S(f_\Theta)] \middle\| \middle\| \mathbb{E}_{\Theta \sim Q}[\mathcal{L}_D(f_\Theta)]\right) \leq \frac{KL(Q||P) + \log \frac{m}{\delta}}{m - 1}$$

*where KL is the Kullback-Leibler (KL) divergence (Kullback & Leibler, 1951).*

Following Dziugaite & Roy (2017), we use the inequality $KL^{-1}(q|c) = \sup\{p \in [0, 1] : KL(q||p) \leq c\} \leq q + \sqrt{c/2}$ to achieve a simple bound on the test error:

$$\mathbb{E}_{\Theta \sim Q}[\mathcal{L}_D(f_\Theta)] \leq KL^{-1}\left(E_{\Theta \sim Q}[\mathcal{L}_S(f_\Theta)] \middle| \frac{KL(Q||P) + \log \frac{m}{\delta}}{m - 1}\right)$$

$$\leq \mathbb{E}_{\Theta \sim Q}[\mathcal{L}_S(f_\Theta)] + \sqrt{\frac{KL(Q||P) + \log \frac{m}{\delta}}{2(m - 1)}}.$$

The intuition from Figure 4 suggests that moving the parameters of each module as close as possible to the initialization value before harming performance. For such $\alpha_i$, we can then define the posterior $Q_i$ for module $i$ to be a Gaussian distribution centered at $\theta_i^\alpha$ with covariance matrix $\sigma_i^2 I$, that is, as if we have additive noise $u_i \sim \mathcal{N}(0, \sigma_i I)$. We use $\Theta^\alpha$ to refer to the case where all $\theta_i$ are replaced with $\theta_i^{\alpha_i}$ and matrix $U$ includes all $u_i$. Then the training loss term can be decomposed as

$$\mathbb{E}_{\Theta \sim Q}[\mathcal{L}_S(f_\Theta)] = \mathbb{E}_{u_i \sim \mathcal{N}(0, \sigma_i I)}[\mathcal{L}_S(f_{\Theta^\alpha + U})]$$

$$\leq \mathcal{L}_S(f_{\Theta^F}) + \left|\mathbb{E}_{u_i \sim \mathcal{N}(0, \sigma_i I)}[\mathcal{L}_S(f_{\Theta^\alpha + U})] - \mathcal{L}_S(f_{\Theta^F})\right|,$$

where the second term on the right hand side of the inequality captures the flatness of the point $\Theta^\alpha$ by adding Gaussian noise and measuring the change in the loss. Therefore, searching over the posterior corresponds to finding a flat solution in the valley that connects the initial and final points. Next, we use this intuition to prove a generalization bound based on module criticality.

First, we express the value of the $\widetilde{\mathcal{O}}(1)$ term in Theorem 3.2 , which is equal to $\epsilon$ as follows:

$$\epsilon = \sum_i \log\left(7m + 2\log\left(\frac{k_i}{k_i \sigma_i^2 + \alpha_i^2 \left\|\theta_i^F - \theta_i^0\right\|_{Fr}^2}\right)\right). \tag{3}$$

Now we proceed with the proof.

The KL-divergence between two $k$-dimensional Gaussian distributions is given by the formula:

$$KL(\mathcal{N}(\mu_1, \Sigma_1)||\mathcal{N}(\mu_P, \Sigma_P)) = \frac{1}{2}\left[\text{tr}\left(\Sigma_2^{-1}\Sigma_1\right) + (\mu_2 - \mu_1)^\top \Sigma_2^{-1}(\mu_2 - \mu_1) - k + \ln(\frac{\det \Sigma_2}{\det \Sigma_1})\right].$$

The above equation can be further simplified for Gaussian distributions with diagonal covariance matrices. Let the prior $P$ be a Gaussian distribution such that for each module $i$, the distribution is $\mathcal{N}(\theta_i^0, \sigma_{P,i}^2 I)$ and let the posterior $Q$ be a Gaussian distribution such that for each module $i$, the distribution is $\mathcal{N}((1 - \alpha)\theta_i^0 + \alpha \theta_i^F), \sigma_{Q,i}^2 I)$. We can then write the KL-divergence $KL(Q||P)$ as

$$KL(Q||P) = \frac{1}{2}\sum_i \left[\frac{k_i \sigma_{Q,i}^2 + \alpha_i^2 \left\|\theta_i^F - \theta_i^0\right\|_{Fr}^2}{\sigma_{P,i}^2} - k_i + k_i \log\left(\frac{\sigma_{P,i}^2}{\sigma_{Q,i}^2}\right)\right]. \tag{4}$$

Since prior should be decided before observing the training set, we are not allowed to optimize for $\sigma_{P,i}$ directly. However, one can optimize for $\sigma_{P,i}$ over a pre-defined set of values and use a union bound argument to get the generalization bound for the best $\sigma_{P,i}$ in that set. We use a covering approach suggested by Langford & Caruana (2002). For $b, \epsilon > 0$, if one chooses the variance of prior to be $\exp(-\epsilon j + b)$ for $j \in \mathbb{N}$ such that for each $j$ the bound holds with probability $1 - \frac{6}{\pi^2 j^2}$, then all bounds hold with probability $1 - \sum_{j \in \mathbb{N}} \frac{6}{\pi^2 j^2} = 1 - \delta$. We can apply the same idea to every module such that the bound holds with probability $1 - \delta \prod_{i=1}^{d} \frac{6}{\pi^2 j_i^2}$.

If we choose $\sigma_{Q,i}^2 \leq 1$ then we have $\sigma_{P,i} \leq \exp\left(\frac{4m}{k_i} + 1\right)$. Otherwise, the bound holds since the right hand side is greater than one. Given (from Equation 3) $\epsilon \geq 0$, if we choose $\sigma_{P,i}^2$ to have the form $\exp\left(\frac{4m - j_i}{k_i} + 1\right)$, for some integer $j_i$, we can always find choose $j_i$ such that

$$k_i \sigma_{Q,i}^2 + \alpha_i^2 \left\|\theta_i^F - \theta_i^0\right\|_{\text{Fr}}^2 \leq k_i \sigma_{P,i}^2 \leq \exp(1/k_i)\left(k_i \sigma_{Q,i}^2 + \alpha_i^2 \left\|\theta_i^F - \theta_i^0\right\|_{\text{Fr}}^2\right). \tag{5}$$

Therefore, the KL-divergence can be bounded as

$$\text{KL}(Q\|P) \leq \frac{1}{2} \sum_i \left[ k_i \frac{k_i \sigma_{Q,i}^2 + \alpha_i^2 \left\|\theta_i^F - \theta_i^0\right\|_{\text{Fr}}^2}{k_i \sigma_{Q,i}^2 + \alpha_i^2 \left\|\theta_i^F - \theta_i^0\right\|_{\text{Fr}}^2} - k_i + k_i \log\left(\frac{\sigma_{P,i}^2}{\sigma_{Q,i}^2}\right) \right]$$

$$= \frac{1}{2} \sum_i \left[ k_i \log\left(\frac{\sigma_{P,i}^2}{\sigma_{Q,i}^2}\right) \right]$$

$$\leq \frac{1}{2} \sum_i k_i \log\left(\frac{\exp(1/k_i)\left(k_i \sigma_{Q,i}^2 + \alpha_i^2 \left\|\theta_i^F - \theta_i^0\right\|_{\text{Fr}}^2\right)}{k_i \sigma_{Q,i}^2}\right)$$

$$\leq \frac{1}{2} \sum_i k_i \log\left(\frac{\exp(1/k_i)\left(k_i \sigma_{Q,i}^2 + \alpha_i^2 \left\|\theta_i^F - \theta_i^0\right\|_{\text{Fr}}^2\right)}{k_i \sigma_{Q,i}^2}\right)$$

$$\leq \frac{1}{2} \sum_i 1 + k_i \log\left(1 + \frac{\alpha_i^2 \left\|\theta_i^F - \theta_i^0\right\|_{\text{Fr}}^2}{k_i \sigma_{Q,i}^2}\right).$$

Note that in order to achieve the inequality in Equation 5, $j_i$ should be chosen as

$$j_i = \left\lfloor \frac{4m}{k_i} + 1 + \log\left(\frac{k_i}{k_i \sigma_{Q,i}^2 + \alpha_i^2 \left\|\theta_i^F - \theta_i^0\right\|_{\text{Fr}}^2}\right) \right\rfloor \leq 5m + \log\left(\frac{k_i}{k_i \sigma_{Q,i}^2 + \alpha_i^2 \left\|\theta_i^F - \theta_i^0\right\|_{\text{Fr}}^2}\right).$$

Given that each such bound should hold with probability $1 - \delta \prod_{i=1}^{d} \frac{6}{\pi^2 j_i^2}$, the log term in the bound can be written as

$$\log \frac{m}{\delta} + \sum_i \log(\pi^2 j_i^2 / 6) \leq \log \frac{m}{\delta} + 2 \sum_i \log\left(7m + 2\log\left(\frac{k_i}{k_i \sigma_{Q,i}^2 + \alpha_i^2 \left\|\theta_i^F - \theta_i^0\right\|_{\text{Fr}}^2}\right)\right).$$

Putting everything together proves the theorem statement.

## B  A DETERMINISTIC GENERALIZATION BOUND FOR CONVOLUTIONAL NETWORKS

We start by stating a generalization bound given by Neyshabur et al. (2018) with a slight improvement in the constants.

**Lemma B.1** (Neyshabur et al. (2018)). *Let $f_\Theta : \mathcal{X} \to \mathbb{R}^C$ be any predictor function with parameters $\Theta$ and $P$ be a prior distribution on parameters $\Theta$. Then for any $\gamma, m, \delta > 0$, with probability $1 - \delta$*

*over the training set $S$ of size $m$, for any parameter $\Theta$ and any perturbation distribution $Q$ over parameters such that $\mathbb{P}_{U \sim Q}\left[\max_{x \in \mathcal{X}} |f_{\Theta+U}(x) - f_{\Theta}(x)| \leq \frac{\gamma}{4}\right] \geq \frac{1}{2}$, we have*

$$\mathcal{L}_D(f_\Theta) \leq \mathcal{L}_{S,\gamma}(f_\Theta) + \sqrt{\frac{2KL(\Theta + U \| P) + 1 + \log \frac{m}{\delta}}{2(m-1)}}.$$

The lemma above gives a data-independent deterministic bound which depends on the maximum change of the output function over the domain after a perturbation. We combine Lemma B.1 with Theorem 3.2 and prove a bound on the perturbation which leads to the following theorem.

**Theorem B.2.** *Let input $x$ be an $N \times N$ image whose norm is bounded by $B$, $f_\Theta : \mathcal{X} \to \mathbb{R}^C$ be the predictor function with parameters $\Theta$ which is a DNN of depth $d$ made of convolutional blocks. Then for any margin $\gamma$, sample size $m$, $\delta > 0$, with probability $1 - \delta$ over the training set $S$, any parameter $\Theta$ and any $\alpha_i > 0$ such that $\max_{x \in \mathcal{X}} |f_\Theta(x) - f_{\Theta^\alpha}(x)| \leq \frac{\gamma}{8}$, we have*

$$\mathcal{L}_D(f_\Theta) \leq \mathcal{L}_{S,\gamma}(f_\Theta) + \sqrt{\frac{\sum_{i=1}^{d} k_i \log\left(1 + \frac{[32edB\alpha_i \|\theta_i^F - \theta_i^0\|_{Fr} \prod_{i \neq j} \|\theta_i^\alpha\|_2 \sqrt{\log(4dN^2)}]^2}{c_i \gamma^2}\right) + \log\left(\frac{m}{\delta}\right) + \widetilde{\mathcal{O}}(1)}{m-1}},$$

(6)

*where $k_i$ is the number of parameters in module $i$. For example, for a convolution module with kernel size $q_i \times q_i$ and number of output channels $c_i$, $k_i = q_i^2 c_{i-1} c_i$.*

*Proof.* First, we express the value of the $\widetilde{\mathcal{O}}(1)$ term in Theorem B.2 , which is equal to $\epsilon_2$ as follows:

$$\epsilon_2 = 1 + \sum_i \log\left(7m + 2\log\left(\frac{k_i}{k_i\gamma^2 / \left(16e \prod_{j \neq i} \|\theta_i^\alpha\|_2 \log(4dN^2)\right)^2 + \alpha_i^2 \|\theta_i^F - \theta_i^0\|_{Fr}^2}\right)\right).$$

(7)

We note that for any $\Theta, \Theta'$, if $\max_{x \in \mathcal{X}} \|f_\Theta(X) - f_\Theta\|_\infty \leq \gamma/2$ then $\mathcal{L}(f_\Theta) \leq \mathcal{L}_\gamma(f'_\Theta)$. The reason is that the output for each class can change by at most $\gamma/2$ and therefore the label can only change for the data points that are within $\gamma$ of the margin.

We start using the assumptions on the perturbation bound. Combining the results from Theorem B.1 and Theorem 3.2, we can get the following bound.

$$\mathcal{L}_D(f_{\Theta^F}) \leq \mathcal{L}_{D,\frac{\gamma}{4}}(f_{\Theta^\alpha})$$

(8)

$$\leq \mathcal{L}_{S,\frac{3\gamma}{4}}(f_{\Theta^\alpha}) + \sqrt{\frac{\frac{1}{2}\sum_{i=1}^{d} k_i \log\left(1 + \frac{\alpha_i^2 \|\theta_i^F - \theta_i^0\|_{Fr}^2}{k_i \sigma_i^2}\right) + \log\frac{m}{\delta} + \epsilon_2}{m-1}}$$

$$\leq \mathcal{L}_{S,\gamma}(f_{\Theta^\alpha}) + \sqrt{\frac{\frac{1}{2}\sum_{i=1}^{d} k_i \log\left(1 + \frac{\alpha_i^2 \|\theta_i^F - \theta_i^0\|_{Fr}^2}{k_i \sigma_i^2}\right) + \log\frac{m}{\delta} + \epsilon_2}{m-1}}$$

where $\epsilon_2$ is given above in Equation 7.

Therefore, it suffices to find the value of $\sigma_i$ under which the assumption on norm of perturbation in function space holds and then simplify the following upper bound given the desired value of $\sigma_i$.

In order to find the desired value of $\sigma_i$ we use the following two lemmas. First we adopt the perturbation lemma by Neyshabur et al. (2018) to bound the change in the output a network based on the magnitude of the perturbation:

**Lemma B.3** (Neyshabur et al. (2018))**.** *Let norm of input $x$ be bounded by $B$. For any $B > 0$, let $f_\Theta : \mathcal{X} \to \mathbb{R}^C$ be a neural network with ReLU activations and depth $d$. Then for any $\Theta, x \in \mathcal{X}$, and any perturbation $U$ s.t. $\|u_i\|_2 \leq \|\theta_i\|_2$, the change in the output of the network can be bounded as follows*

$$\|f_{\Theta+U} - f_\Theta\|_2 \leq eB \prod_{i=1}^{d} \|\theta_i\|_2 \sum_{j=1}^{d} \frac{\|u_i\|_2}{\|\theta_i\|_2}.$$

(9)

We next use the following lemma by Pitas et al. (2017) that bounds the magnitude of the Gaussian perturbation $u_i$ for each convolutional module based on the standard deviation of the perturbation.

**Lemma B.4** (Pitas et al. (2017)). *Let $u_i$ be a Gaussian perturbation for each module $i$ of a convolutional model. Let $N$ be the image size, $q_i$, $c_i$ be the kernel size and the number of output channels at module $i$ respectively. We have that*

$$\mathbb{P}\left[\|u_i\|_2 \geq \sigma_i\big(q_i(2\sqrt{c_i}) + t\big)\right] \leq 2N^2 e^{-\frac{t^2}{2q_i^2}}.$$

The lemma above suggests that by taking union bounds over all modules, we can ensure that with probability $1/2$ we have that for any module $i$, the following upper bound on the spectral norm of the perturbation holds.

$$\|u_i\|_2 \leq \sigma_i q_i(2\sqrt{c_i} + \sqrt{2\log(4dN^2)}) \leq 2\sigma_i q_i(\sqrt{c_i} + \sqrt{\log(4dN^2)}) \leq 4\sigma_i q_i\sqrt{c_i \log(4dN^2)}.$$

Combining this with perturbation bound in Equation 9, we have that

$$\|f_{\Theta^\alpha + U} - f_{\Theta^\alpha}\|_2 \leq eB\sum_{i=1}^d \|u_i\|_2 \prod_{j\neq i}^d \|\theta_j^\alpha\|_2 \leq 4eB\sum_{i=1}^d \sigma_i q_i\sqrt{c_i \log(4dN^2)} \prod_{j\neq i}^d \|\theta_i^\alpha\|_2 \leq \frac{\gamma}{8},$$

where the last inequality can be achieved with

$$\sigma_i = \frac{\gamma}{32edB\prod_{i=1}^d \|\theta_i^\alpha\|_2\, q_i\sqrt{c_i \log(4dN^2)}}. \tag{10}$$

Therefore, this value for $\sigma_i$ ensures the assumption on norm of perturbation in function space in Theorem B.2 holds and hence completes the proof.

Moreover, we show how we get the value of $\epsilon_2$, by showing the simplification from inserting the value for $\sigma_i$ from Equation 10 as follows.

$$\log\left(1 + \frac{\alpha_i^2\left\|\theta_i^F - \theta_i^0\right\|_{\text{Fr}}^2}{k_i\sigma_i^2}\right) \leq \log\left(1 + \frac{\left[\alpha_i\left\|\theta_i^F - \theta_i^0\right\|_{\text{Fr}}\right]^2}{q_i^2 c_i^2 \sigma_i^2}\right)$$

$$\leq \log\left(1 + \frac{[32edB\alpha_i\left\|\theta_i^F - \theta_i^0\right\|_{\text{Fr}} \prod_{i=1}^d \|\theta_i^\alpha\|_2\, q_i\sqrt{c_i \log(4dN^2)}]^2}{q_i^2 c_i^2 \gamma^2}\right)$$

$$= \log\left(1 + \frac{[32edB\alpha_i\left\|\theta_i^F - \theta_i^0\right\|_{\text{Fr}} \prod_{i=1}^d \|\theta_i^\alpha\|_2\, \sqrt{\log(4dN^2)}]^2}{c_i \gamma^2}\right).$$

Then, $\epsilon_2$ can also be simplified as follows.

$$\epsilon_2 = 1 + \sum_i \log\left(7m + 2\log\left(\frac{k_i}{k_i\sigma_i^2 + \alpha_i^2\left\|\theta_i^F - \theta_i^0\right\|_{\text{Fr}}^2}\right)\right)$$

$$\leq 1 + \sum_i \log\left(7m + 2\log\left(\frac{k_i}{k_i\gamma^2/\left(32edB\prod_{i=1}^d \|\theta_i^\alpha\|_2\, q_i\sqrt{c_i \log(4dN^2)}\right)^2 + \alpha_i^2\left\|\theta_i^F - \theta_i^0\right\|_{\text{Fr}}^2}\right)\right)$$

$$\leq 1 + \sum_i \log\left(7m + 2\log\left(\frac{k_i}{\gamma^2/\left(32edB\prod_{i=1}^d \|\theta_i^\alpha\|_2\, \sqrt{\log(4dN^2)}\right)^2 + \alpha_i^2\left\|\theta_i^F - \theta_i^0\right\|_{\text{Fr}}^2}\right)\right)$$

$$\leq 1 + \sum_i \log\left(7m + 2\log(k_i) - 4\log\left(\alpha_i\left\|\theta_i^F - \theta_i^0\right\|_{\text{Fr}} + \gamma/32edB\prod_{i=1}^d \|\theta_i^\alpha\|_2\, \sqrt{\log(4dN^2)}\right)\right).$$

$\square$

## C  DETAILS ON EXPERIMENTAL SET-UP

For all our experiments, we use the CIFAR10 and CIFAR100 datasets. To train our networks we used Stochastic Gradient Descent (SGD) with momentum 0.9 to minimize multi-class cross-entropy loss. On CIFAR10 each model is trained until the cross-entropy loss on the training dataset falls below 0.19. While on CIFAR100 each model is trained until the cross-entropy loss on the training dataset falls below 0.25. The ResNets, DenseNets and VGGs were trained using a stage-wise constant learning rate scheduling with a starting learning rate of 0.1 and with a decrease by a multiplicative factor of 0.2 every 60 epochs. FCN was trained with an initial learning rate of 0.1 with a decrease by a multiplicative factor of 0.2 every 200 epochs. Batch size of 128 was used for all models and weight decay with factor 5e-4 was used to train all networks.

We mainly study three types of neural network architectures:

- Fully Connected Networks (FCNs): The FCNs consist of 2 fully connected layers. FCN (I) contains 5000 and 1000 hidden units respectively while FCN (II) contains 10000 and 2000 hidden units respectively. Each of these hidden layers is followed by a batch normalization layer and a ReLU activation. The final output layer (that follows the ReLU activation in the second layer) has an output dimension of 10 or 100 (number of classes).

- VGGs: Architectures by Simonyan & Zisserman (2015) that consists of multiple convolutional layers, followed by multiple fully connected layers and a final classifier layer (with output dimension 10 or 100). We study the VGG with 11 and 16 layers.

- DenseNets: Architectures by Huang et al. (2017) that consists of multiple convolutional layers, followed by a final classifier layer (with output dimension 10 or 100). We study the DenseNet with 121 layers.

- ResNets: Architectures used are ResNets V1 (He et al., 2016). All convolutional layers (except downsample convolutional layers) have kernel size $3 \times 3$ with stride 1. Downsample convolutions have stride 2. All the ResNets have five stages (0-4) where each stage has multiple residual/downsample blocks. These stages are followed by a maxpool layer and a final linear layer. Here are further details about the ResNets used in the paper:

    - ResNet18: ResNet18 architechtures studied in the paper have 1 convolutional layer in Stage 0 (64 ouput channels), Stage 1 has 2 residual blocks (64 output channels), Stage 2 has one downsample block and one residual block (128 output channels), Stage 3 has one downsample block and one residual block (256 output channels) and Stage 4 again has one downsample block and a residual block (512 output channels).

    - ResNet34: ResNet34 architectures in this paper have 5 stages. Stage 0 has 1 convolutional layer with 64 output channels followed by a ReLU activation. Stage 1 has 3 residual blocks (64 output channels), Stage 2 has 1 downsample block and 3 residual blocks (128 output channels), Stage 3 has 1 downsample block and 5 residual blocks (256 output channels) and, Stage 4 has 1 downsample block and 2 residual blocks (512 output channels).

    - ResNet50: ResNet50 architectures in this paper again have 5 stages. Stage 0 has 1 convolutional layer with 64 output channels followed by a ReLU activation. Stage 1 has 1 downsample block and 2 residual blocks (256 output channels), Stage 2 has 1 downsample block and 3 residual blocks (512 output channels), Stage 3 has 1 downsample block and 5 residual blocks (1024 output channels) and, Stage 4 has 1 downsample block and 2 residual blocks (2048 output channels).

    - ResNet101: ResNet101 architectures in this paper again have 5 stages. Stage 0 has 1 convolutional layer with 64 output channels followed by a ReLU activation. Stage 1 has 1 downsample block and 2 residual blocks (256 output channels), Stage 2 has 1 downsample block and 3 residual blocks (512 output channels), Stage 3 has 1 downsample block and 22 residual blocks (1024 output channels) and, Stage 4 has 1 downsample block and 2 residual blocks (2048 output channels).

The ResNets, DenseNets and VGGs in the paper are trained without batch normalization.

During training, images are padded with 4 pixels of zeros on all sides, then randomly flipped (horizontally) and cropped. Global mean and standard deviation are computed on all training images

and applied to normalize the inputs. While training a ResNet18 on the CIFAR10 dataset with 20% of the labels randomly corrupted, we *do not* augment the training set with images that are randomly flipped and cropped. We also do not use weight decay during training these networks.

# D   ADDITIONAL EXPERIMENTS

In these set of experiments we compare the generalization performance of four ResNet18 architectures where we vary the number of output channels in each stage. In the ResNet18 (1x width) network the number of output channels are 16, 16, 32, 64, 128 in the five stages respectively. The other ResNet18s have their output channels scaled by factors of 2,4 and 8 in each stage. Table 4 summarizes our results for networks trained on the CIFAR10 dataset and Table 5 summarizes are results for networks trained on the CIFAR100 dataset.

We find that separating these networks based on a complexity measure is a much more challenging as these four networks differ by just the channel widths at the different stages. We find that on this task *all complexity measures* that we studied (including ours) does poorly. It is an interesting question for future research to see if our complexity measure can be refined to separate these networks as well.

Table 4: Measuring complexity of different architectures trained on CIFAR10. The ResNet18 architectures have different channel widths. The network ResNet18 (1x width) has 16,16,32,64,128 channels in the five stages.

| ResNet18 (x) | GE | PFN | PSN | DtI | NoP | SoSP | PAC Bayes | Net. Criticality |
|---|---|---|---|---|---|---|---|---|
| 1x width | 6.27% | 4e17 | 1e12 | 1409 | 6.9e5 | 6.0e7 | 3.0e5 | 1.0e5 |
| 2x width | 5.13% | 8e19 | 4e13 | 2253 | 2.7e6 | 2.9e8 | 4.2e5 | 1.3e5 |
| 4x width | 4.61% | 1e22 | 4e14 | 3430 | 1.1e7 | 1.3e9 | 6.9e5 | 2.2e5 |
| 8x width | 2.88% | 1e24 | 2e15 | 5365 | 4.4e7 | 5.6e9 | 2.7e6 | 6.7e5 |
| Kendall's $\tau$ | - | -1 | -1 | -1 | -1 | -1 | -1 | -1 |

Table 5: Measuring complexity of different architectures trained on CIFAR100. The ResNet18 architectures have different channel widths. The network ResNet18 (1x width) has 16,16,32,64,128 channels in the five stages.

| ResNet18 (x) | GE | PFN | PSN | DtI | NoP | SoSP | PAC Bayes | Net. Criticality |
|---|---|---|---|---|---|---|---|---|
| 1x width | 30.4% | 6e18 | 3e12 | 2650 | 7.1e5 | 7.3e7 | 3.2e6 | 1.5e6 |
| 2x width | 31.8% | 1e21 | 4e13 | 4248 | 2.8e6 | 3.4e8 | 2.7e6 | 1.3e6 |
| 4x width | 30.6% | 2e22 | 9e14 | 4855 | 1.1e7 | 1.4e9 | 3.4e6 | 1.6e6 |
| 8x width | 28.4% | 3e25 | 1e18 | 9269 | 4.4e7 | 8.0e9 | 6.9e6 | 2.8e6 |
| Kendall's $\tau$ | - | -0.33 | -0.33 | -0.33 | -0.33 | -0.33 | -0.66 | -0.66 |

# E    FIGURES

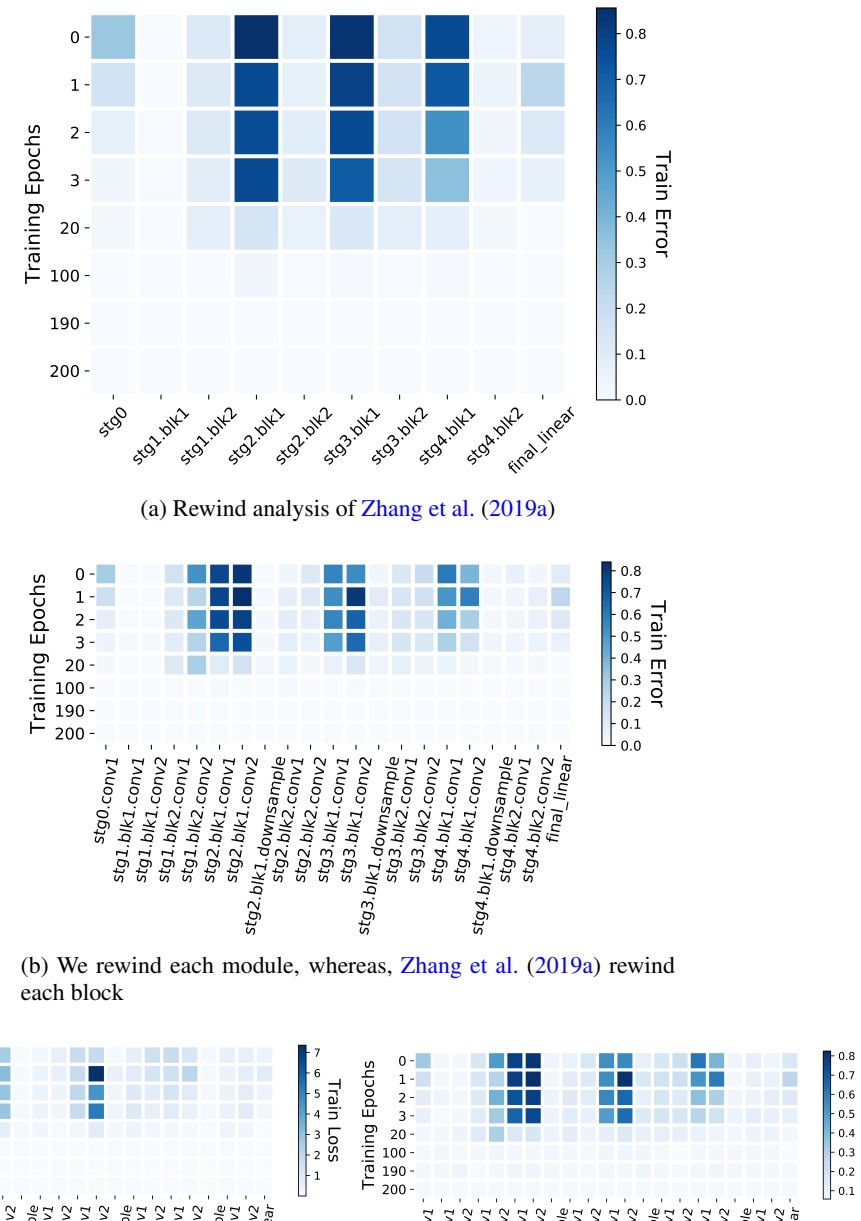

(a) Rewind analysis of Zhang et al. (2019a)

(b) We rewind each module, whereas, Zhang et al. (2019a) rewind each block

(c) The effect of rewinding on train loss

(d) The effect of rewinding on test error

Figure 6: Analysis of rewinding modules to initialization for the ResNet-18 architecture. Each row represents a layer in ResNet18-v1 and each column represents a particular training epoch that the module is rewound to.

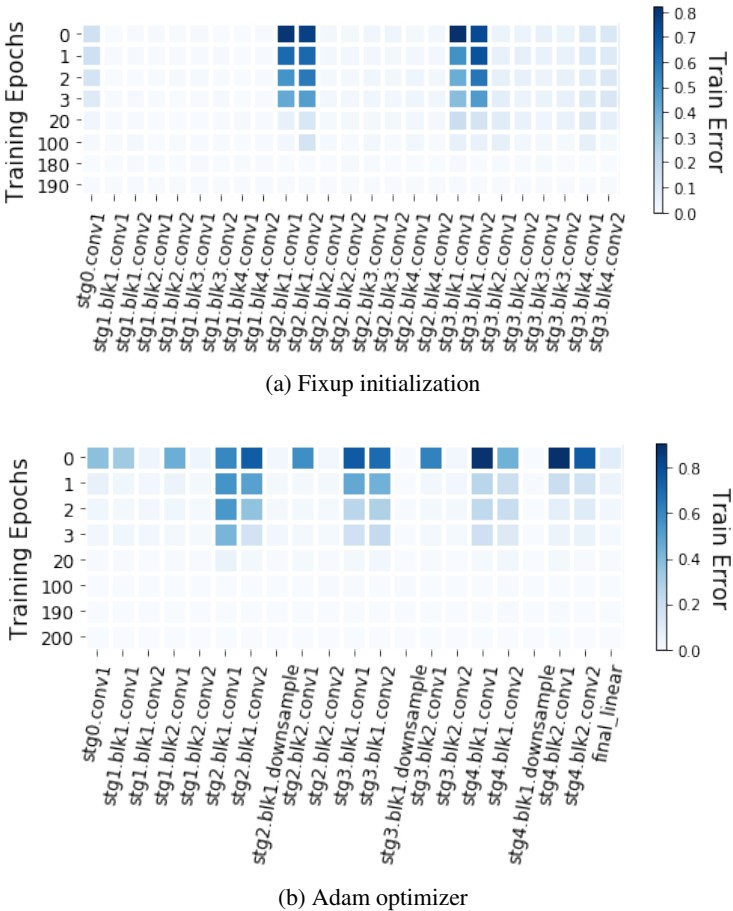

(a) Fixup initialization

(b) Adam optimizer

Figure 7: Criticality pattern of Resnet18 when trained with Fixup initialization and with the Adam optimizer

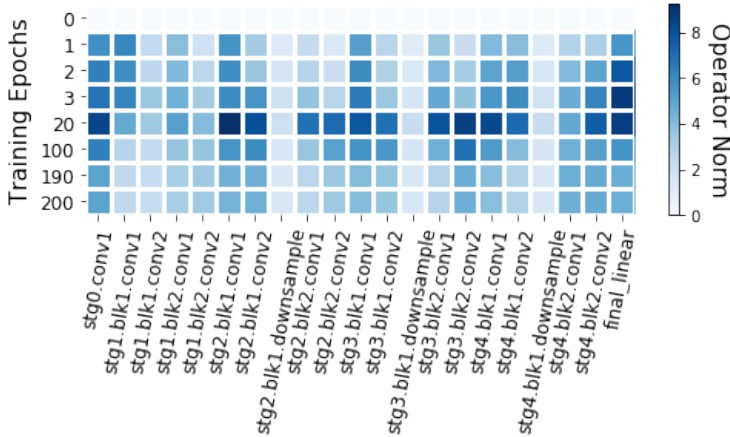

Figure 8: Operator norm of difference from initialization

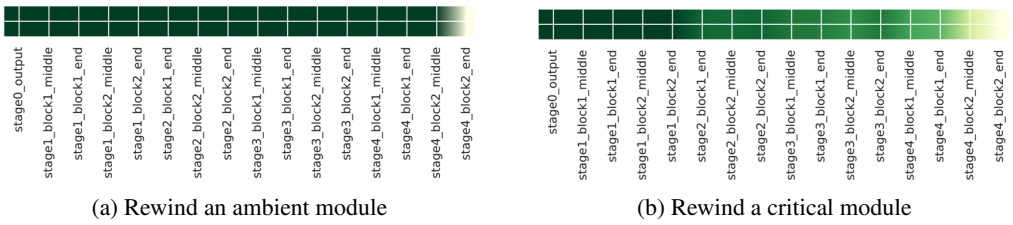

(a) Rewind an ambient module         (b) Rewind a critical module

Figure 9: Similarity in activation patterns when an ambient or critical module is rewound. Darker green denotes higher similarity

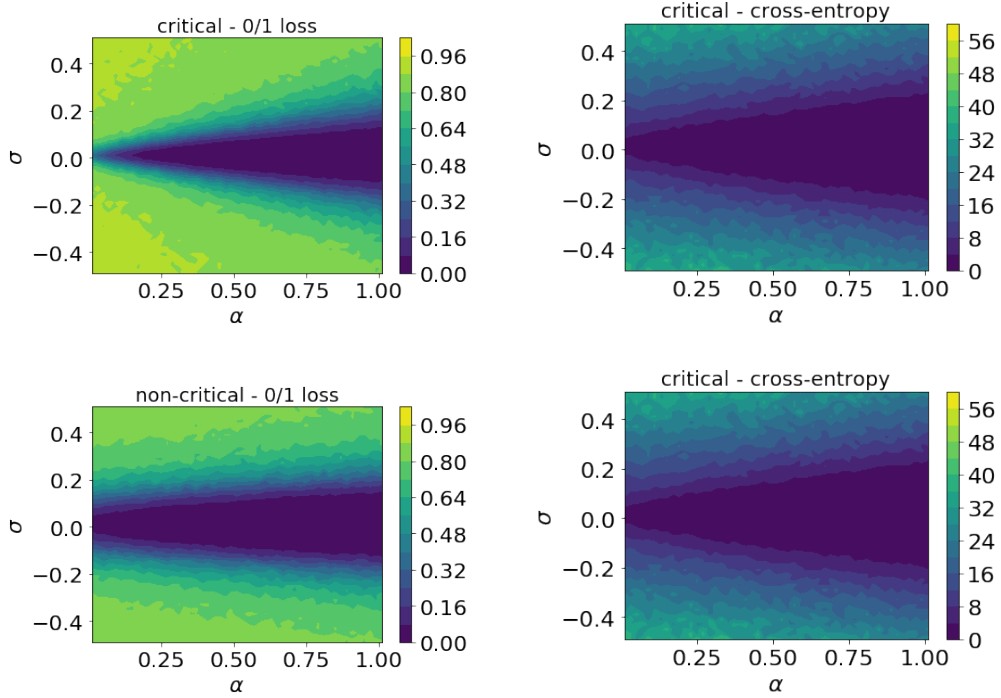

Figure 10: 0/1 loss and cross-entropy loss for critical and non-critical modules, for given different values of $\sigma$ and $\alpha$ in Definition 3.1.

