# OpenReview forum: "The intriguing role of module criticality in the generalization of deep networks"
_ICLR.cc/2020/Conference — Accept (Spotlight)_

### Official Review · AnonReviewer3 · 2019-10-20
**Official Blind Review #3**

**Rating:** 6

**Review:**

The paper builds upon the "module criticality" phenomenon and proposes a quantitative approach to measure this at the module and the network level.  A module's criticality is low if when it is switched to its initialization value, the error does not change drastically.

The paper uses a convex combination of the initial weights and the final weights of a layer/module to define an optimization path to traverse. The authors quantitatively define the module criticality such that it depends on how much closer the weights can get to the initial weights on this path while still being robust to random permutations. The network critically is defined as the sum of the module criticality measure of all the layers.

Empirical results on CIFAR10 show that the network's criticality is reflective of the generalization performance. For example, increasing resnet depth leads to improved generalization and low criticality. Though intuitively, it is not clear why moving closer to the initial values and thus lower average criticality indicated better generalization. It will be useful ot have a discussion on this issue. Results on other datasets will also be useful.

Overall, the network criticality measure appears a useful tool to predict the generalization performance compared to other measures such as distance from initialization, weight spectrum, and others.

**Experience Assessment:**

I have read many papers in this area.

**Review Assessment: Checking Correctness Of Derivations And Theory:**

I assessed the sensibility of the derivations and theory.

**Review Assessment: Checking Correctness Of Experiments:**

I assessed the sensibility of the experiments.

**Review Assessment: Thoroughness In Paper Reading:**

I made a quick assessment of this paper.

---

> ### Author Response · Authors · 2019-11-12
> **Author response**
>
> Thank you for your positive comments on the paper. We have updated the paper to address both of your suggestions adequately and we hope you consider increasing your score if you find these changes satisfactory.
>
> - Intuitively moving closer to the initialization values would indicate that the effective function class is smaller and hence the network should generalize better. For example, in the extreme case where none of the weights change from their initialization value the function class would be a single function (the initial function) and the generalization error would be very low (~0%) as both the train and test error would be very high (but equal).
>
> - Thank you for your suggestion. We have added 3 networks: ResNet50, VGG11 and another fully connected network to our experimental section. We have also repeated our experiments on the CIFAR100 dataset.

---

> > ### Author Response · Authors · 2019-11-14
> > **Added Densenet experiments.**
> >
> > Apart from the changes listed above we have also updated our submission to include DenseNets in our experiments.

---

### Official Review · AnonReviewer2 · 2019-10-23
**Official Blind Review #2**

**Rating:** 8

**Review:**

This paper introduces a new way to reason about neural network generalization using a module criticality measure. The measure is tangible and intuitive. It leads to some formal bounds on the generalization of deep networks, and is able to better rank trained image classification architectures than previous measures.

I am leaning to accept, as I expect this to be a significant theoretical contribution with several potential practical applications. With a few additional details, this could be a very strong submission:

(1)	Choice of module decomposition. Having each module be a single convolutional or fully-connected layer makes intuitive sense, but is there some theoretical motivation for this choice? If the only requirement for a module is that it includes some linear transformation, in the extreme, a module could consist of a single weight, or the entire network. Would those choices change the generalization bounds or relative criticality across different architectures?
(2)	Scope of experimental results. The ranking results would be much more compelling if they included a broader range of architectures, including more recent models with more branching, e.g., DenseNet. Is there some reason ResNet101 has higher generalization error than 18 and 34? Net. Criticality for ResNets is inversely correlated with the number of layers; is there an explanation for this? Is this true for other very deep models?
(3)	Practical use. To compute the criticality measure, we must train the model; but, if we train the model, we can compute generalization directly. So, what is the practical application of the measure? Is there some way it could be used to save computation? Could it help in the case of a small validation dataset, which we do not want to look at many times during model selection?

Minor typos:
-	Section 2.2: “An stable phenomena”
-	Section 2.3: “…an the…”
-	In appendix: “ResNet101: ResNet34 architectures…”

----------------------------

After rebuttal:

The authors have addressed my concerns, and I've increased my rating. There are still a few points I'd like to see addressed in the final version:

1. The fact that the approach cannot yet be applied to batch normalization is a big practical drawback. Some discussion of approaches you tried, why they didn't work, and possible future directions for overcoming this would be appreciated.

2. Clarify in the paper that the "PAC Bayes" approach used for comparison (Table 1) is a your method, i.e., an ablated version of criticality. As is, someone reading the paper quickly may think all you've done is add an alpha parameter to an existing "PAC Bayes" approach, which does fairly well on its own.

3. Visualizing the experimental tables as scatterplots could make them easier for a reader to interpret.




**Experience Assessment:**

I have read many papers in this area.

**Review Assessment: Checking Correctness Of Derivations And Theory:**

I assessed the sensibility of the derivations and theory.

**Review Assessment: Checking Correctness Of Experiments:**

I assessed the sensibility of the experiments.

**Review Assessment: Thoroughness In Paper Reading:**

I read the paper at least twice and used my best judgement in assessing the paper.

---

> ### Author Response · Authors · 2019-11-12
> **Author response**
>
> Thank you for your very encouraging remarks and useful feedback. We have added the discussions and experiments you suggested (and even more) to the paper. In light of this and your strong review, we hope you would consider increasing your score to “accept”.
>
>
> We next address your questions in detail:
> (1) Thanks for this insightful question! As you correctly pointed out, the choice of module decomposition is somewhat arbitrary. One could choose a module to comprise of a single scalar weight or the entire network and this would lead to different generalization scores. To answer your question more directly, the theoretical results hold for any decomposition. Why did we choose the modules this way? Since we are looking at a linear combination of parameters with their initialization, it makes sense to choose modules to be transformations that are linear in the parameters. Otherwise, the output would be very sensitive to a linear combination with initialization. Among all such decompositions, we chose the one with the minimum number of modules. In a sense, modules are chosen to be the largest well-behaved units in the network (and that happens to be the most natural choice). It is certainly possible that a smarter choice for module exists for each architecture which could lead to a lower network criticality measure. We are encouraged by your question and will add more discussion around this issue to the paper.
>
> (2) Thank you for your suggestions. We have added 3 networks, ResNet50, VGG11 and another fully connected network, to our experimental section. We have also repeated all our experiments on the CIFAR100 dataset. Moreover, we have  added another complexity metric to our table of comparison. Therefore, our current empirical results are much stronger than the submitted version.
>
>
> The network criticality for ResNets is inversely correlated on the CIFAR10 dataset, but this is not true in the CIFAR100 dataset (which we just added). Here ResNet101 has the lowest generalization error and that is reflected by the network criticality score. We believe that the reason ResNet101 has higher generalization error for CIFAR10 in our experiments is that we do not use batch normalization to train. We make this choice since it is not clear how to accurately rewind batch normalization layers (we have tried several natural choices and they did not work).
>
> (3) Since criticality measure correlates with generalization, we would ideally want to encourage this measure to be low. This can happen in two different ways: 1) We can design regularizers based on the criticality measure and add them to the objective function. This would ensure that the trained network has low criticality. 2) We can explicitly design architecture with low criticality measure. This for example can happen by adding an explicit change in the modules (such as rewinding them to their initialization) to make sure the the learned network has low criticality. Another potential application is architecture search. As you mentioned, the model selection is usually done using a validation set but the search is over a large number of models, the result would overfit to the validation set. Therefore, measures such as the criticality score can be used in these scenarios.
>
> We have corrected the typos that you identified in our updated draft. Thanks for pointing them out!

---

> > ### Author Response · Authors · 2019-11-14
> > **Added Densenet experiments.**
> >
> > Apart from the changes listed above we have also updated our submission to include DenseNets in our experiments as you suggested.

---

### Official Review · AnonReviewer4 · 2019-11-08
**Official Blind Review #4**

**Rating:** 6

**Review:**

The paper introduces concept of "module criticality" to understand the role played by several modules in a model and how this affects the generalization of the models. This is quite an important problem to study as this helps to develop better understanding of the current architectures and potentially reduce their size without suffering the accuracy drop. This is a great theoretical contribution.

The paper studies this per module compared to previous works where the entire architecture is rewounded. The paper also studies this for ResNet models which are more widely/practically used than just the fully connected layers alone. This helps better understand the model as a whole.

The authors do a good robust experimental study for different network initialization, various CNN models like ResNet18, 34, 101, VGG16 and also FCN. The results demonstrate the module criticality to be a good metric for generalization of models.

Overall, a good paper.

**Experience Assessment:**

I have read many papers in this area.

**Review Assessment: Checking Correctness Of Derivations And Theory:**

I did not assess the derivations or theory.

**Review Assessment: Checking Correctness Of Experiments:**

I assessed the sensibility of the experiments.

**Review Assessment: Thoroughness In Paper Reading:**

I made a quick assessment of this paper.

---

> ### Author Response · Authors · 2019-11-12
> **Author response**
>
> Thank you for your positive remarks about both our theoretical contributions and our experimental study. We are a bit puzzled that your positive remarks are not reflected in the final score (weak reject). We hope that “weak reject” is chosen by mistake; otherwise, we would be happy to answer any concerns/questions you may have about our submission.

---

### Public Comment · ~Cemal_Gurpnar1 · 2020-09-04
**Architecture Selection**

First of all, I would like to thank you for your grateful work. My question is about architecture selection. If we want to use network criticality measure for an architecture selection, should I train the architectures that I try to make a selection in on the data and after that make a selection or can I use the network criticality measures you found in your paper for any dataset?

If I summarize my question, are the network criticality measures you found in your paper for architectures like ResNet 18, VGG 16 etc. generic and can be usable for any dataset?

Thank you.

---

### Decision · Program_Chairs · 2019-12-19

**Decision:**

Accept (Spotlight)

**Comment:**

The paper analyses the importance of different DNN modules for generalization performance, explaining why certain architectures may be much better performing than others. All reviewers agree that this is an interesting paper with a novel and important contribution.